# Accurate Uncertainty Estimation and Decomposition in Ensemble Learning

**Jeremiah Zhe Liu**[*]
Google Research & Harvard University
zhl112@mail.harvard.edu

**John Paisley**
Columbia University
jpaisley@columbia.edu

**Marianthi-Anna Kioumourtzoglou**
Columbia University
mk3961@cumc.columbia.edu

**Brent A. Coull**
Harvard University
bcoull@hsph.harvard.edu

## Abstract

Ensemble learning is a standard approach to building machine learning systems that capture complex phenomena in real-world data. An important aspect of these systems is the complete and valid quantification of model uncertainty. We introduce a *Bayesian nonparametric ensemble* (BNE) approach that augments an existing ensemble model to account for different sources of model uncertainty. BNE augments a model's prediction and distribution functions using Bayesian nonparametric machinery. It has a theoretical guarantee in that it robustly estimates the uncertainty patterns in the data distribution, and can decompose its overall predictive uncertainty into distinct components that are due to different sources of noise and error. We show that our method achieves accurate uncertainty estimates under complex observational noise, and illustrate its real-world utility in terms of uncertainty decomposition and model bias detection for an ensemble in predict air pollution exposures in Eastern Massachusetts, USA.

## 1 Introduction

Ensemble learning has a long history in areas such as robust engineering system design [4], financial investment management [20], and weather and climate forecasting [35], where high-risk decisions and critical projections are made in the presence of noise and uncertainty. Failure to accurately quantify the predictive uncertainty in these ensemble systems can lead to severe consequences [1], such as the market crash of 2008.

To properly quantify predictive uncertainty, it is important for an ensemble learning system to recognize different types of uncertainties that arise in the modeling process. In machine learning modeling, two distinct types of uncertainties exist: *aleatoric uncertainty* and *epistemic uncertainty* [22] (see Figure 1). Aleatoric uncertainty arises due to the stochastic variability inherent in the data generating process, for example due to an imperfect sensor, and is described by the cumulative distribution function (CDF) $F(y|\mathbf{x}, \Theta)$ of the data specified by a given model. On the other hand, epistemic uncertainty arises due to our lack of knowledge about the data generating mechanism. A model's epistemic

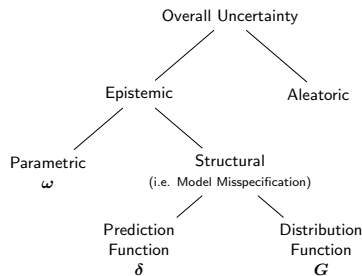

Figure 1: A decomposition of different types of uncertainty by the Bayesian Nonparametric Ensemble (BNE).

---

[*]Work done at Harvard University.

uncertainty can be reduced by collecting more data, whereas aleatoric uncertainty is irreducible since it is inherent to the data generating mechanism. A machine learning model's epistemic uncertainty can arise from two sources [42]: *parametric uncertainty* that reflects uncertainty associated with estimating the model parameters under the current model specification, which can be described by a Bayesian model's posterior $p(\Theta|y, \mathbf{x})$; and *structural uncertainty* that reflects the uncertainty about whether a given model specification is sufficient for describing the data, i.e. whether there exists a systematic discrepancy between CDF $F(y|\mathbf{x}, \Theta)$ based on the model and the data-generating distribution $F^*(y|\mathbf{x})$.

The goal of uncertainty estimation is to properly characterize both a model's aleatoric and epistemic uncertainties [24, 42]. In regions that are well represented by the training data, a model's aleatoric uncertainty should accurately estimate the data-generating distribution by flexibly capturing the stochastic pattern in the data (i.e., calibration [19]), while in regions unexplored by the training data, the model's epistemic uncertainty should increase to capture the model's lack of confidence in the resulting predictions (i.e. out-of-distribution generalization [24]). Within the epistemic uncertainty, the structural uncertainty needs to be estimated to identify the sources of structural biases in the ensemble model, and to quantify how these structural biases may impact the model output, something necessary for the continuous model validation and refinement of a running ensemble system [40, 34].

A comprehensive framework for quantifying these three types of uncertainties is currently lacking in the ensemble learning literature. We refer readers to Supplementary Section A for a full review and how our work is related to existing literature. Briefly, existing methods typically handle the aleatoric uncertainty using an assumed distribution family (e.g., Gaussian) [24, 48] that may not capture the stochastic patterns in the data (e.g. asymmetry, heavy-tailedness, multimodality, or their combinations). Work exists on quantifying epistemic uncertainty, although ensemble methods mainly work with collections of base models of the same class, and usually do not explicitly characterize the model's structural uncertainty [6, 9, 10, 50, 24, 51, 27].

In this work, we develop an ensemble model that addresses all three sources of predictive uncertainty. Our specific contributions are: 1) We propose *Bayesian Nonparametric Ensemble* (BNE), an augmentation framework that mitigates misspecification in the original ensemble model and flexibly quantifies all three sources of predictive uncertainty (Section 2). 2) We establish BNE's model properties in uncertainty characterization, including its theoretical guarantee with respect to consistent estimation of aleatoric uncertainty, and its ability to decompose different sources of epistemic uncertainties (Section 3). 3) We demonstrate through experiments that the proposed method achieves accurate uncertainty estimation under complex observational noise and improves predictive accuracy (Section 4), and illustrate our method by predicting ambient fine particle pollution in Eastern Massachusetts, USA by ensembling three different existing prediction models developed by multiple research groups (Section 5).

## 2 Bayesian Nonparametric Ensemble

In this section, we introduce the Bayesian Nonparametric Ensemble (BNE), an augmentation framework for ensemble learning. We focus on the application of BNE to regression tasks. Given an ensemble model, BNE mitigates the original model's misspecification in the prediction function and in the distribution function using Bayesian nonparametric machinery. As a result, BNE enables an ensemble to flexibly quantify aleatoric uncertainty in the data, and account for both the parametric and the structural uncertainties.

We build the full BNE model by starting from the classic ensemble model. Denoting $F^*(y|\mathbf{x})$ the CDF of data-generating distribution for an continuous outcome. Given an observation pair $\{\mathbf{x}, y\} \in \mathbb{R}^p \times \mathbb{R}$ where $y \sim F^*(y|\mathbf{x})$ and a set of base model predictors $\{f_k\}_{k=1}^K$, a classic ensemble model assumes the form

$$Y = \sum_{k=1}^K f_k(\mathbf{x}) \omega_k + \varepsilon, \tag{1}$$

where $\omega = \{\omega_k\}_{k=1}^K$ are the ensemble weights assigned to each base model, and $\varepsilon$ is a random variable describing the distribution of the outcome. For simplicity of exposition, in the rest of this section we assume $\omega$ and $\varepsilon$ follow independent Gaussian priors, which corresponds to a classic stacking model assuming a Gaussian outcome [10].

In practice, given a set of predictors $\{f_k\}_{k=1}^K$'s built by domain experts, a practitioner needs to first specify a distribution family for $\varepsilon$ (e.g. Gaussian such that $\varepsilon \sim N(0, \sigma_\varepsilon)$), then estimate $\omega$ and $\varepsilon$ using collected data. During this process, two types of model biases can arise: *bias in prediction function* $\mu = \sum_{k=1}^K f_k(\mathbf{x})\omega_k$ caused by the systematic bias shared among all the base predictors $f_k$'s; and *bias in distribution specification* caused by assuming a distribution family for $\varepsilon$ that fails to capture the stochastic pattern in the data, producing inaccurate estimates of aleatoric uncertainty. BNE mitigates these two types of biases that exist in (1) using Bayesian nonparametric machinery.

**Mitigate prediction bias using residual process $\delta$**    To mitigate model's structural bias in prediction, BNE first adds to (1) a flexible residual process $\delta(\mathbf{x})$, so the ensemble model becomes a semiparametric model [11, 39]:

$$Y = \sum_{k=1}^K f_k(\mathbf{x})\omega_k + \delta(\mathbf{x}) + \varepsilon. \tag{2}$$

In this work, we model $\delta(\mathbf{x})$ nonparametrically using a Gaussian process (GP) with zero mean function $\mathbf{0}(x) = 0$ and kernel function $k_\delta(\mathbf{x}, \mathbf{x}')$. The residual process $\delta(\mathbf{x})$ adds additional flexibility of the model's mean function $E(Y|\mathbf{x})$, and domain experts can select a flexible kernel for $\delta$ to best approximate the data-generating function of interest (e.g., a RBF kernel to approximate arbitrary continuous functions over a compact support [33]). As a result, in densely-sampled regions that are well captured by the training data, $\delta(\mathbf{x})$ will confidently mitigate the prediction bias between the observation $y$ and the prediction function $\sum_{k=1}^K f_k(\mathbf{x})\omega_k$. However, in sparsely-sampled regions, the posterior mean of $\delta(\mathbf{x})$ will be shrunk back towards $\mathbf{0}(x) = 0$, so as to leave the predictions of the original ensemble (1) intact (since these expert-built base models presumably have been specially designed for the problem being considered) and the posterior uncertainty of $\delta(\mathbf{x})$ will be larger to reflect the model's increased structural uncertainty in its prediction function at location $\mathbf{x}$.

We recommend selecting $k_\delta$ from the shift-invariant kernel family $k(\mathbf{x}, \mathbf{x}') = g(\mathbf{x} - \mathbf{x}')$. Shift-invariant kernels are well suited for characterizing a model's epistemic uncertainty, since the resulting predictive variances are explicitly characterized by the distance from the training data, which yields predictive uncertainty that increases as the prediction location of interest is farther away from data [36].

We write the model CDF of (2) as $\Phi_\varepsilon(y|\mathbf{x}, \mu)$. In the case $\varepsilon \sim N(0, \sigma_\varepsilon^2)$, $\Phi_\varepsilon$ is a Gaussian CDF with mean $\mu$ and variance $\sigma_\varepsilon^2$. Notice that since $\delta(\mathbf{x})$ is a Gaussian process, (2) specifies $Y$ as a hierarchical Gaussian process with mean function $\sum_{k=1}^K f_k(\mathbf{x})\omega_k$ and kernel function $k_\delta(\mathbf{x}, \mathbf{x}') + \sigma_\varepsilon^2$.

**Mitigate distribution bias using calibration function $G$**    Although flexible in its mean prediction, the model in (2) can still be restrictive in its distributional assumptions. That is, at a given location $\mathbf{x} \in \mathbb{R}^p$, because the model corresponds to a Gaussian process specification for $Y$, the posterior of (2) still follows a Gaussian distribution [36]. Consequently, when the data distribution is multimodal, non-symmetric, or heavy-tailed, the model in (2) can still fail to capture the underlying data-generating distribution $F^*(y|\mathbf{x})$, resulting in systematic discrepancy between $\Phi_\varepsilon(y|\mathbf{x}, \mu)$ and $F^*(y|\mathbf{x})$.

To mitigate this bias in the specification of the data distribution, BNE further augments $\Phi_\varepsilon(y|\mathbf{x}, \mu)$ by using a nonparametric function $G$ to "calibrate" the model's distributional assumption using observed data $\mathbf{z} = \{y, \mathbf{x}\}$, i.e., BNE models its CDF as $F(y|\mathbf{x}, \mu) = G[\Phi_\varepsilon(y|\mathbf{x}, \mu)]$. As a result, the full BNE model's CDF is a flexible nonparametric function capable of modeling a wide range of complex distributions. In this work, we model $G$ using a Gaussian process with identity mean function $I(x) = x$ and kernel function $k_G$, and we impose probit-based likelihood constraints on $G$ so it respects the mathematical property of a CDF (i.e. monotonic and bounded between $[0, 1]$, see Section B for detail). As a result, the full BNE model's CDF follows a *constrained Gaussian process* (CGP) [29, 30, 38]:

$$F(y|\mathbf{x}, \mu) \sim CGP\Big(\Phi_\varepsilon(y|\mathbf{x}, \mu), k_G(\mathbf{z}, \mathbf{z}')\Big), \tag{3}$$

where $\mathbf{z} = \{y, \mathbf{x}\}$. In this work, we set $k_G$ to the Matérn $\frac{3}{2}$ kernel $k_{\text{Matérn } 3/2}(d) = (1 + \sqrt{3}d/l) * exp(-\sqrt{3}d/l)$ where $d = ||\mathbf{x} - \mathbf{x}'||_2$. The sample space of a Matérn $\frac{3}{2}$ Gaussian process corresponds to the space of Hölder continuous functions that are at least once differentiable, allowing $F$ to flexibly model the space of (Lipschitz) continuous CDFs $F^*(y|\mathbf{x})$ whose probability density function (PDF) exist [46]. Consequently, in regions well represented by the training data, the BNE's model CDF will

flexibly capture the complex patterns in the data distribution. In regions outside the training data, the BNE's model CDF will fall back to $\Phi_\varepsilon(y|\mathbf{x}, \mu)$, not interfering with the generalization behavior of the original ensemble model. Additionally, the posterior uncertainty in (3) will reflect the model's additional structural uncertainty with respect to its distribution specification.

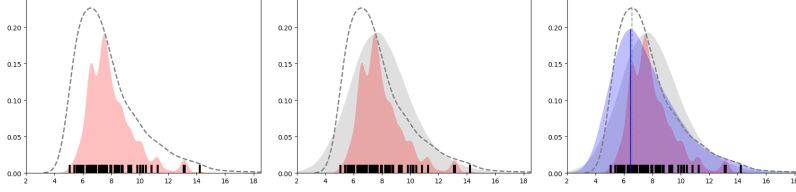

Figure 2: Illustrative example illustrating impact of $G$ on model's posterior predictive distribution. Dashed Line: True Distribution $F^*(y|\mathbf{x})$, Black Ticks: Observations, Red Shade: predictive density of $\sum_k f_k \omega_k$, Grey Shade: predictive density of $\Phi_\varepsilon(y|\mathbf{x}, \mu)$ (Gaussian assumption), Blue Shade: predictive density of $G \circ \Phi_\varepsilon(y|\mathbf{x}, \mu)$ (nonparametric noise correction).

To further illustrate the role $G$ plays in the BNE's ability to flexibly characterize an outcome distribution, we consider an illustrative example where we run the BNE model both with and without $G$ to predict $y$ at a fixed location of $\mathbf{x}$ (i.e. estimating the conditional distribution $F^*(y|\mathbf{x})$ at fixed location of $\mathbf{x}$) where $y|\mathbf{x} \sim Gamma(1.5, 2)$ (Figure 2). As shown, the posterior distribution of $\Phi_\varepsilon(y|\mathbf{x}, \mu)$ (grey shade) fails to capture the skewness in the data's empirical distribution, and consequently yields a biased maximum a posterior (MAP) estimate due to its restrictive distributional assumptions. On the other hand, the full BNE model $F(y|\mathbf{x}, \mu) = G[\Phi_\varepsilon(y|\mathbf{x}, \mu)]$ is able to calibrate its predictive distribution (blue shade) toward the data distribution using $G$, and consequently produces improved characterization of $F^*(y|\mathbf{x})$ and improved MAP estimate.

**Model Summary** To recap, given a classic ensemble model (1), BNE nonparametrically augments the model's prediction function with a residual process $\delta$, and augments the model's distribution function with a calibration function $G$. Specifically, for data $y|\mathbf{x}$ that is generated from the distribution $F^*(y|\mathbf{x})$, the full BNE assumes the following model:

$$F^*(y|\mathbf{x}) = G[\Phi_\varepsilon(y|\mathbf{x}, \mu)], \quad \mu = \sum_{k=1}^{K} f_k(\mathbf{x})\omega_k + \delta(\mathbf{x}). \tag{4}$$

The priors are defined to be

$$G \sim CGP(I, k_G), \quad \delta \sim GP(\mathbf{0}, k_\delta), \quad \omega \sim N(0, \sigma_\omega^2 \mathbf{I}),$$

where $k_G$ is the Matérn $\frac{3}{2}$ kernel, and $k_\delta$ is a shift-invariant kernel to be chosen by the domain expert (we set it to Matérn $\frac{3}{2}$ in this work). The zero-mean GP ensures the ensemble bias term $\delta$ reverts to zero out of sample, while the identity-mean GP allows the noise process to be white Gaussian noise $\varepsilon$ out of sample. In other words, this prior structure allows BNE to flexibly capture data distribution where data exists, and revert to the classic ensemble otherwise.

BNE's hyper-parameters are the Matérn length-scale parameters $l_\delta$ and $l_G$, and the prior variances $\sigma_\omega$ and $\sigma_\varepsilon$. Consistent with the existing GP approaches, we place the inverse-Gamma priors on the $l_\delta$ and $l_G$ and the Half Normal priors on $\sigma_\omega$ and $\sigma_\varepsilon$ [43]. Posterior sampling is performed using Hamiltonian Monte Carlo (HMC) [2], for which we pre-orthogonalize kernel matrices with respect to their mean functions to avoid parameter non-identifiability [31, 37]. The time complexity for sampling from the BNE posterior is $O(N^3)$ due to the need to invert the $N \times N$ kernel matrices. For large datasets, we can consider the parallel MCMC scheme proposed in [26] which partitions the data into $K$ subsets and estimates the predictive intervals with reduced complexity $O(N^3/K^2)$. Section C describes posterior inference in further detail.

## 3 Characterizing Model Uncertainties with BNE

### 3.1 Mitigating Model Bias under Uncertainty

In this section we study the contribution of BNE's model components to an ensemble's prediction and predictive uncertainty estimation. For a model with predictive CDF $F(y|\mathbf{x})$, we notice that

the model's predictive behavior is completely characterized by $F(y|\mathbf{x})$: a model's predictive mean is expressed as $E(y|\mathbf{x}) = \int_{y \in \mathbb{R}} [I(y > 0) - F(y|\mathbf{x})] dy$, and a model's $(1 - q)\%$ predictive interval is expressed as $U_q(y|\mathbf{x}) = [F^{-1}(1 - \frac{q}{2}|\mathbf{x}), F^{-1}(1 + \frac{q}{2}|\mathbf{x})]$ [12]. Consequently, BNE improves upon an ensemble model's prediction and uncertainty estimation by building a flexible model for $F$ that better captures the data-generating $F^*(y|\mathbf{x})$.

**Bias Correction for Prediction and Uncertainty Estimation**   We can express the predictive mean of BNE as:

$$E(y|\mathbf{x}, \omega, \delta, G) = \sum_{k=1}^{K} f_k(\mathbf{x})\omega_k + \underbrace{\delta(\mathbf{x})}_{\text{due to } \delta} + \underbrace{\int_{y \in \mathscr{Y}} \left[ \Phi(y|\mathbf{x}, \mu) - G[\Phi(y|\mathbf{x}, \mu)] \right] dy}_{\text{due to } G}. \quad (5)$$

See Supplementary E for derivation. As shown, the predictive mean for the full BNE is composed of three parts: 1) the predictive mean of the original ensemble $\sum_{k=1}^{K} f_k(\mathbf{x})\omega_k$; 2) the term $\delta$ representing BNE's "direct correction" to the prediction function; and 3) the term $\int \left[ \Phi(y|\mathbf{x}, \mu) - G[\Phi(y|\mathbf{x}, \mu)] \right] dy$ representing BNE's "indirect correction" to prediction obtained upon the relaxation of the original Gaussian assumption in model CDF. We denote these two error-correction terms as $D_\delta(y|\mathbf{x})$ and $D_G(y|\mathbf{x})$.

To express BNE's estimated predictive uncertainty, we denote as $\Phi_{\varepsilon, \omega}$ the predictive CDF of the original ensemble (1) (i.e. with mean $\sum_k f_k \omega_k$ and variance $\sigma_\varepsilon^2$). Then BNE's predictive interval is:

$$U_q(y|\mathbf{x}, \omega, \delta, G) = \left[ \Phi_{\varepsilon, \omega}^{-1} \left( G^{-1}(1 - \frac{q}{2}|\mathbf{x}) \right) + \delta(\mathbf{x}), \ \Phi_{\varepsilon, \omega}^{-1} \left( G^{-1}(1 + \frac{q}{2}|\mathbf{x}) \right) + \delta(\mathbf{x}) \right]. \quad (6)$$

 Comparing (6) to the predictive interval of original ensemble $[\Phi_{\varepsilon, \omega}^{-1}(1 - \frac{q}{2}), \Phi_{\varepsilon, \omega}^{-1}(1 + \frac{q}{2})]$, we see that the locations of the BNE predictive interval endpoints are adjusted by the residual process $\delta$, while the spread of the predictive interval (i.e. the predictive uncertainty) is calibrated by $G$.

**Quantifying Uncertainty in Bias Correction**   A salient feature of BNE is that it can quantify its uncertainty in bias correction. This is because the bias correction terms $D_\delta$ and $D_G$ are random quantities that have posterior distributions (since they are functions of $\delta$ and $G$). Specifically, we can quantify the posterior uncertainty in whether $D_\delta$ and $D_G$ are different from zero by estimating $P(D_\delta(y|\mathbf{x}) > 0)$ and $P(D_G(y|\mathbf{x}) > 0)$, i.e., the percentiles of 0 in the posterior distribution of $D_\delta$ and $D_G$. Values close to 0 or 1 indicate strong evidence that model bias impacts model prediction. Values close to 0.5 indicate a lack of evidence of this impact, since the posterior distributions of these error-correction terms are roughly centered around zero. This approach can be generalized to describe the impact of the distribution biases on other properties of the predictive distribution (e.g. skewness, multi-modality, etc. see Section E for detail).

### 3.2   Consistent Estimation of Aleatoric Uncertainty

Recall that a model characterize the aleatoric uncertainty in data through its model CDF. As it is clear from the expression of predictive interval $U_q(y|\mathbf{x}) = [F^{-1}(1 - \frac{q}{2}|\mathbf{x}), \ F^{-1}(1 + \frac{q}{2}|\mathbf{x})]$, for a model to reliably estimate its predictive uncertainty, the model CDF $F$ should be estimated to be consistent with the data-generating CDF $F^*(y|\mathbf{x})$, such that, for example, the 95% predictive interval $U_{0.95}(y|\mathbf{x})$ indeed contains the observations $y \sim F^*(y|\mathbf{x})$ 95% of the time. This consistency property is known in the probabilistic forecast literature as *calibration* [19], and defines a mathematically rigorous condition for a model to achieve reliable estimation of its predictive uncertainty. To this end, using the flexible calibration function $G$, BNE enables its model CDF to consistently capturing the data-generating $F^*(y|\mathbf{x})$:

**Theorem 1** (Posterior Consistency). *Let $F = G[\Phi]$ be a realization of the CGP prior defined in (3). Suppose that the true data-generating CDF $F^*(y|\mathbf{x})$ is contained in the support of $F$. Given $\{y_i, \mathbf{x}_i\}_{i=1}^{n}$, a random sample from $F^*(y|\mathbf{x})$, denote the expectation with respect to $F^*$ as $E^*$ and denote the posterior distribution as $\Pi_n$. There exists a sequence $\varepsilon_n \to 0$ and sufficiently large $M$ such that*

$$E^* \Pi_n \left( ||F^* - F||_2 \geq M\varepsilon_n \Big| \{y_i, \mathbf{x}_i\}_{i=1}^{n} \right) \to 0.$$

We defer the full proof to Section D. This result states that, as the sample size grows, the BNE's posterior distribution of $F$ concentrates around the true data-generating CDF $F^*$, therefore consistently capture the aleatoric uncertainty in the data distribution. By setting $k_G$ to the Matérn $\frac{3}{2}$ kernel, the prior support of BNE is large and contains the space of compactly supported, Lipschitz continuous $F^*$'s whose PDF exist [5, 46]. The convergence speed of the posterior $F$ depends both on the distance of $F^*$ relative to the prior distribution, and on how close the smoothness of the Matérn prior matches the smoothness of $F^*$ [44, 45]. To this end, the BNE improves its speed of convergence by centering $F$'s prior mean to $\Phi(y|\mathbf{x}, \omega)$ and by estimating the kernel hyperparameter $l_G$ adaptively through an inverse Gamma prior.

### 3.3 Uncertainty Decomposition

For an ensemble model that is augmented by BNE, the goal of uncertainty decomposition is to understand how different sources of uncertainty combine to impact the ensemble model's predictive distribution, and to distinguish the contribution of each source in driving the overall predictive uncertainty. As shown in Figure 1, the posterior uncertainty in each of a BNE's model parameters $\{\omega, \delta, G\}$ accounts for an important source of model uncertainty. Consequently, both the aleatoric and epistemic uncertainties are quantified by the BNE's posterior distribution, and can be distinguished through a careful decomposition of the model posterior.

We first show how to separate the aleatoric and epistemic uncertainties in BNE's posterior predictive distribution. Consistent with existing approaches, we use *entropy* to measure the overall uncertainty in a model's predictive distribution: $\mathscr{H}(y|\mathbf{x}, \theta) = -\int_{y \in \mathscr{Y}} f(y|\theta) * \log f(y|\theta) dy$ [18, 32]. Entropy measures the average amount of information contained in a distribution, and is reduced to a function of variance when the distribution is Gaussian. Given a posterior distribution of the model parameters $p(\omega, \delta, G)$, we separate the aleatoric and epistemic uncertainties in the ensemble model's predictive distribution $p(y|\mathbf{x}) = \int f(y|\mathbf{x}, G, \delta, \omega) dp(\omega, \delta, G)$ as [15]:

$$\mathscr{H}(y|\mathbf{x}) = \underbrace{\mathscr{I}\big((\omega, \delta, G), y|\mathbf{x}\big)}_{epistemic} + \underbrace{E_{G, \delta, \omega}\Big[\mathscr{H}(y|\mathbf{x}, G, \delta, \omega)\Big]}_{aleatoric}, \tag{7}$$

where the second term measures the model's aleatoric uncertainty (i.e. which describes the noise patterns inherence to $y$) by computing the expected entropy coming from the model distribution $f(y|\mathbf{x}, G, \delta, \omega)$ that is averaged over the model's posterior belief about $\{G, \delta, \omega\}$. The first term is the *mutual information* between $p(\omega, \delta, G)$ and $p(y|\mathbf{x})$, and measures a model's overall epistemic uncertainty (both parametric and structural) encoded in the joint posterior $p(\omega, \delta, G)$ [14, 18].

We now show how to separate the overall epistemic uncertainty $\mathscr{I}((\omega, \delta, G), y|\mathbf{x})$ into its parametric and structural components. This further decomposition is important in understanding how the ensemble model's predictive uncertainty changes by accounting for the fact that its prediction and distribution functions may be misspecified. Specifically, recall that an ensemble model's parametric uncertainty is the uncertainty about the ensemble weights under the *current model specification* (i.e. by assuming $\delta = 0, G = I$). Therefore the model's parametric uncertainty is encoded in the conditional posterior $p(\omega|\delta = 0, G = I)$ and can be measured by the conditional mutual information $\mathscr{I}(\omega, y|\mathbf{x}, \delta = 0, G = I)$. The model's structural uncertainty contains two components: (1) uncertainty about the prediction function (accounted by $\delta$) and (2) uncertainty about the distribution function (accounted by $G$). The first component describes the model's additional uncertainty about $\omega$ and $\delta$ *under current distribution assumption* (i.e. by assuming $G = I$), which is encoded in the difference between $p(\omega, \delta|G = I)$ and $p(\omega|\delta = 0, G = I)$. The second component describes the model's additional uncertainty about $\omega$, $\delta$ and $G$ by relaxing also the distribution assumption, which is encoded in the difference between $p(\omega, \delta, G)$ and $p(\omega, \delta|G = I)$. By measuring these additional uncertainties using differences between mutual information, we decompose the overall epistemic uncertainty as:

$$\mathscr{I}\big((\omega, \delta, G), y|\mathbf{x}\big) = \underbrace{\mathscr{I}((\omega, \delta, G), y|\mathbf{x}) - \mathscr{I}((\omega, \delta), y|\mathbf{x}, G = I)}_{structural, G} +$$
$$\underbrace{\mathscr{I}((\omega, \delta), y|\mathbf{x}, G = I) - \mathscr{I}(\omega, y|\mathbf{x}, \delta = 0, G = I)}_{structural, \delta} + \underbrace{\mathscr{I}(\omega, y|\mathbf{x}, \delta = 0, G = I)}_{parametric}.$$

where $\mathscr{I}(\theta, y|\theta') = \int f(\theta, y|\theta') \log \frac{f(\theta, y|\theta')}{f(\theta|\theta') f(y|\theta')} d\theta dy$ denotes the conditional mutual information. All three uncertainty terms in the above expression are non-negative (see Section F.1). Computing

these uncertainty terms is straightforward since under BNE, $p(\omega|\delta = 0, G = I)$ and $p(\omega, \delta|G = I)$ both have closed form, which correspond to the posterior of (1) and (2), respectively. We present an example of such a decomposition in the air pollution application (Section 5).

## 4 Experiments

This section reports an in-depth validation of the proposed method on a nonlinear function approximation task with complex (heterogeneous and heavy-tailed) observation noise. We illustrate the method's ability in uncertainty decomposition and bias detection by visualizing the decomposition of model's predictive distribution into their aleatoric, parametric, and structural components, and also visualize the impact of model bias to the model's output distribution using method described in Section 3.1. We then interrogate the method's operating characteristics in prediction (RMSE to true $E^*(y|\mathbf{x})$) and uncertainty quantification ($L_1$ distance to true $F^*(y|\mathbf{x})$), and these metrics' convergence behavior with respect to the increasing sample sizes. We consider a time series problem with heterosdecastic noise with varying degree of skewness in $P(y|\mathbf{x})$ and with imbalanced sampling probability in $x$ (see Figure 3). The detailed experiment settings are documented in Supplementary G.

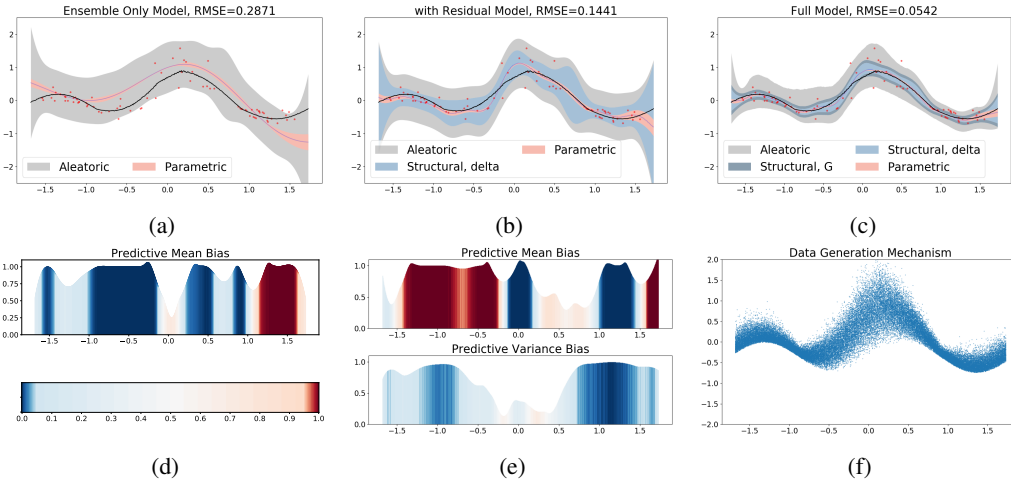

Figure 3: **First Column**: **(a)** Uncertainty decomposition in the original ensemble; **(d)** Posterior confidence in 3a's bias in predictive mean due to prediction function misspecification. **Second Column**: **(b)** Uncertainty decomposition in the BNE model without $G$; **(e)** Posterior confidence in 3b's bias in predictive mean and variance due to distribution misspecification. **Third Column**: **(c)** Uncertainty decomposition in the full BNE model; **(f)** Data generation mechanism. Blue Line: Posterior Mean Prediction. Shaded Region: 90% posterior credible intervals for model's parametric, structural and aleatoric uncertainties.

**Uncertainty Quantification and Decomposition** Figure 3 visually illustrates the role each model component play in BNE's ability in prediction (predictive mean) and uncertainty quantification (90% predictive intervals), and furthermore, how the structural uncertainty encoded in $\delta$ and $G$ is used to diagnose the impact of model bias on its predictive distribution. We run the original Bayesian ensemble (i.e. BNE without $\delta$ and $G$), the **Bayesian Additive Ensemble** (BAE) (i.e. BNE without $G$), and the full BNE model on 100 data points (red dots in Figure 3a-3e). As shown, the original ensemble model (Figure 3a), restricted by its parametric assumption, produces a predictive distribution that fail to capture observations even in the training set. The BAE (Figure 3b) improves the original ensemble by mitigating the systematic bias in prediction, and help the model to better account for its epistemic uncertainty by increasing the predictive uncertainty at locations where data is scarce. However, BAE's aleatoric uncertainty is still biased in that it is roughly constant throughout the range of $x$, failing to account for the heterogeneity in observation's variance, a pattern that is evident in data's empirical distribution. Finally, the full BNE model (Figure 3c) flexibly transforms its predictive distribution to better capture the empirical distribution of the data. As a result, it is able to properly account for the heterogeneity in the observation noise, and at the same time produced improved prediction. Figure 3d and 3e quantify the impact of original ensemble's model bias on model's predictive mean and variances (see Section G for further description).

**Operating Characteristics** We benchmark BNE against its abalated version (**BAE**) and also classic and recent nonparametric and ensemble methods: the classic **Kernel Conditional Distribution Estimator** (CondKDE) that fits the conditional distribution nonparametrically using kernel estimators with cross-validated bandwidth [28]. The **Bayesian Mixture of Experts** (BME) combines the predictive distributions adaptively using softmax-transformed Gaussian weights as $\sum_k \pi_k(\mathbf{x})\phi(y|f_k, \sigma_k)$ [50]. The recent **Bayesian Stacking** (stack) [51] which uses non-adaptive weights $\sum_k \pi_k \phi(y|f_k, \sigma_k)$ but calibrates $\pi_k$ using leave-one-out cross validation, and finally the **Deep Ensemble** (DeepEns), which fits a mixture of Gaussians parametrized using neural networks [24]. We vary sample size between 100 and 1000, and repeat the simulation 50 times in each setting. Figure 4 shows the results. We first observe that the patterns of change in RMSE and $L_1$ distance are similar. This is due to the fact that $E(y|\mathbf{x}) = \int[I(y > 0) - F(y|\mathbf{x})]dy$, a model's improvement in estimating $F$ is reflected directly in the improvement in RMSE. As shown, the RMSE and $L_1$ distance for both **stack** and **BAE** stabilized at higher values due to their lack of flexibility in capturing the heterogeneity in the data, producing biased model estimates even in large sample. On the other hand, the mixture-of-Gaussian estimators (BME and DeepEns) and nonparametric estimators (CondKDE and BNE) continuously improve due to the flexibility in their distribution assumptions. Comparing between the best performing models (DeepEns, CondKDE and BNE), we notice that DeepEns has worse generalization performance in small samples, likely due to the instability of neural network estimators in the low data regime. The performance for BNE and CondKDE are comparable in this time series experiment. However we note that it is usually difficult to generalize kernel density estimators to higher dimensions [41].

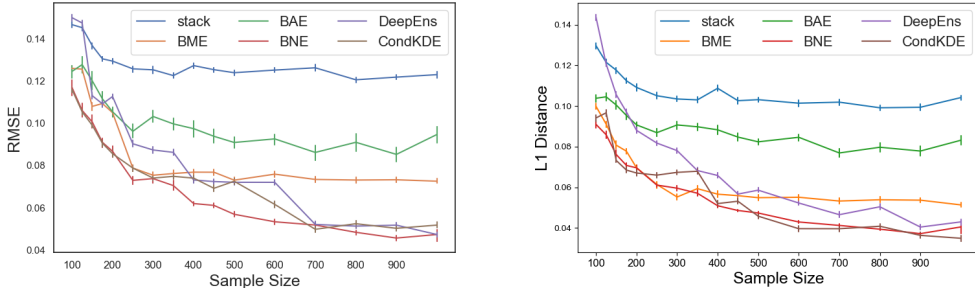

Figure 4: Model's convergence behavior in prediction and uncertainty estimation with respect to $F^*(y|\mathbf{x})$. **Left**: RMSE. **Right**: $L1$ distance $L(F, F^*) = \sum_{\mathbf{x} \in \mathscr{X}} \int |F(y|\mathbf{x}) - F^*(y|\mathbf{x})|dy$.

## 5 Application: Spatial integration of air pollution models in Massachusetts

In this section, we apply BNE to a real-world air pollution prediction ensemble system in Eastern Massachusetts consisted of three state-of-the-art $PM_{2.5}$ exposure models ([23, 16, 47]). We introduce the background of air pollution ensemble systems in Section H. Our goals are to understand the driving factors behind the ensemble system's uncertainty, and detect the ensemble model's systematic bias in predicting annual air pollution concentrations. We implement our ensemble framework on the base models' out-of-sample predictions at 43 monitors in Eastern Massachusetts in 2011.

Figure 5 visualizes the BNE's posterior predictions and uncertainty decomposition across the study region. Further results are summarized in Section H. As shown, due to the sparsity in monitoring locations (only 43 in this modeling area), the model's overall uncertainty is driven mainly by the two types of epistemic uncertainties. More specifically, the model's parametric uncertainty in 5(c) highlights spatial regions where the disagreement in base model predictions has substantial influence on the overall model uncertainty (e.g., the regions northwest to the City of Boston), suggesting further investigations of the performance of individual model predictions in these regions. Further, BNE's posterior estimates in the base models' systematic bias, i.e. $P(D_\delta(y|\mathbf{x}) > 0)$, suggests evidence of over-estimated $PM_{2.5}$ concentrations slightly north of Boston by the coast, and also around a monitor west from Boston, in Worcester, MA (see Supplementary Figure H.3).

## 6 Discussion and Future Work

We developed a principled Bayesian nonparametric augmentation framework for ensemble learning to: 1) mitigate model bias in the prediction and distribution function, and 2) account for model

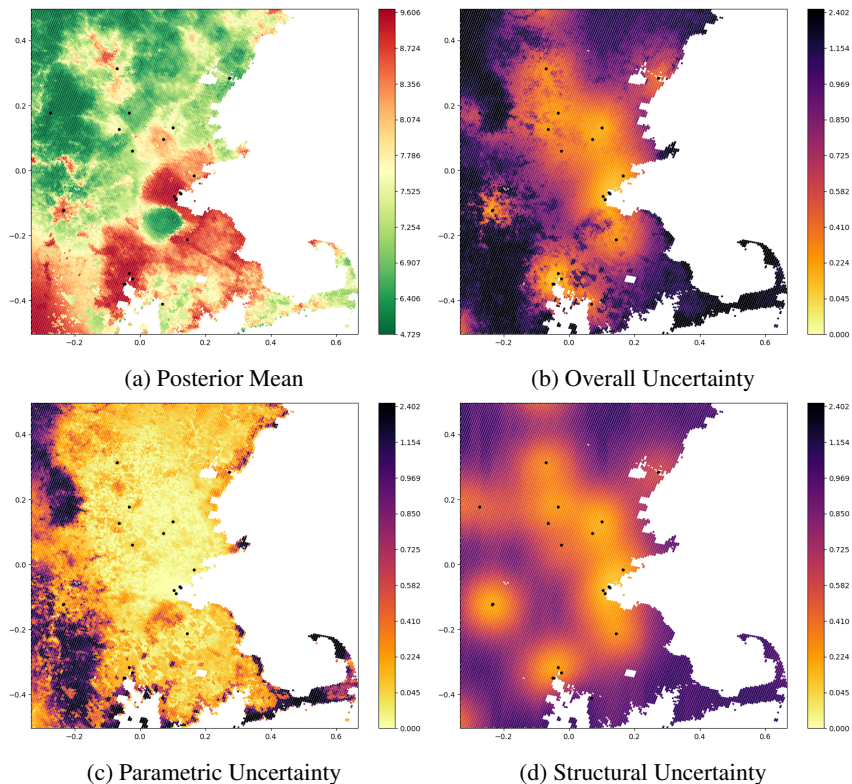

| | | |
|---|---|---|
| (a) Posterior Mean | (b) Overall Uncertainty | |
| (c) Parametric Uncertainty | (d) Structural Uncertainty | |

Figure 5: Posterior Mean and Uncertainty Decomposition in the BNE model.

uncertainties from different sources (aleatoric, parametric, structural) for a continuous outcome with complex observational noise. The main features of this method are accurate estimation of the aleatoric uncertainty, and principled detection and quantification of model misspecification in terms of its impact on model prediction. Experiments showed that the method produces well-calibrated estimation of aleatoric uncertainty and improved prediction under complex observational noise, and also a complete quantification of different sources of epistemic uncertainty. Application to a real-world air pollution prediction problem shows how this method can help in understanding the factors driving model uncertainty, and in detecting the systematic errors in the ensemble system.

There are three important future directions for this work. The first direction is to adapt the BNE framework developed here to high dimension scenarios. This can be achieved by choosing kernel functions for $\delta$ and $G$ that are suitable for high-dimensional problems. Example choices include the additive kernel [17] or (deep) neural network kernel [3, 25]. Alternatively, one could also build variable selection into the model using shrinkage priors such as the Automatic Relevance Determination (ARD), spike-and-slab, or Horseshoe [7, 49]. The second direction is to develop inference algorithm for BNE that are scalable to large dataset and at the same time produces rigorous uncertainty estimates. This is difficult with traditional variational inference algorithms since they usually does not enjoys a guarantee in fully capturing the posterior distribution. The third direction is to develop methods to model other important sources of uncertainty (e.g. algorithmic [8, 13, 21] and data uncertainty [14, 32]) and to quantify their impact on model prediction.

**Acknowledgement** Authors would like to thank Lorenzo Trippa, Jeff Miller, Boyu Ren at Harvard Biostatistics, Yoon Kim at Harvard CS and Ge Liu at MIT EECS for the insightful comments and fruitful discussion. This publication was made possible by USEPA grant RD-83587201. Its contents are solely the responsibility of the grantee and do not necessarily represent the official views of the USEPA. Further, USEPA does not endorse the purchase of any commercial products or services mentioned in the publication. Funding was also provided by NIH grants ES030616 and ES000002.

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
