[Supplementary Material · supplementary.pdf]

# Supplementary for
# Accurate Uncertainty Estimation and Decomposition in Ensemble Learning

## Contents

# A  Discussion and Related Work

Uncertainty quantification (UQ) has a long history in robust engineering system design [4] , financial investment management [23], and weather and climate forecast [47], where high-risk decisions and critical projections are made in the presence of noise and uncertainty. The exact goals of UQ, while varying slightly by discipline, generally fall under one of the two categories: (i) **Accurate Uncertainty Estimation** to properly characterize a model's aleatoric and epistemic uncertainties for the purpose of reliable prediction, and (ii) **Uncertainty / Model Error Decomposition** to identify the sources of model error and uncertainty in the current system, for the purpose of uncertainty reduction and model improvement. We review existing work under these two categories in the following two subsections. Ensemble learning is becoming a standard modeling technique in many of these areas, with the type of base model predictors $\{f_k\}_{k=1}^K$ ranging from deterministic (e.g., rule-based expert systems [10]) to probablistic (e.g. multi-model ensemble of perturbed physics simulations [55, 20, 52]). Although our framework in (1) assumes deterministic base models $f_k$, it can be readily adapted to probablistic predictors by assuming $\varepsilon$ to be the weighted mixture of the probablistic component of the base models.

## A.1  Uncertainty Estimation

The goal of uncertainty estimation is to properly characterize both model's aleatoric and epistemic uncertainties. Specifically, in regions that are well represented by the training data, a model's aleatoric uncertainty should acurately capture the data's empirical distribution (i.e., calibration [22]), while in regions unexplored by the training data, the model's epistemic uncertainty should elevate naturally to capture model's lack of confidence in prediction (i.e., out-of-distribution generalization [31]) .

**Estimating Epistemic Uncertainty** Ensemble methods are often used as a technique to estimate the parametric uncertainty [7**?** ] and to mitigate the structure uncertainty [8, 46] of its base models. However, there exists less work in quantifying the epistemic uncertainty of an ensemble model itself. To this end, there are frequentist approach in deriving a model's asymptotic distribution [38], Bayesian approaches that place structural prior on the model parameters [9, 27, 5, 58, 33, 60], and Monte Carlo methods that perturb model parameters randomly and propagate such parameter uncertainty forward in time through dynamic models [55, 20, 52]. Our work distinguishes from previous literature in that it uses Bayesian nonparametric methods to quantify the epistemic uncertainty in an ensemble model, with an special focus on quantifying the structural uncertainty.

Compare to other approaches, Bayesian nonparametric methods (e.g. Gaussian process) are especially well suited for quantifying the epistemic uncertainty of a model, since its predictive variance is explicitly characterized by the distance from the training data [49]. Specifically, our approach in quantifying a model's structural uncertainty is inspired by two class of model calibration techniques: the *computer model calibration* [26] which calibrates a model's sysmatic component using an Gaussian process, and the *probablistic calibration* [43, 29, 54] which calibrates a model's random component (e.g. the predictive CDF) using regression. Our method subsumes previous work by unifying these two empirically successful practices into one coherent Bayesian model, which enables end-to-end inference through a rigourous treatment of the modeling framework, and allows principled quantification of structural uncertainty by performing Bayesian inference [3]. Furthermore, we give explicit consideration to the model's out-of-distribution behavior in uncertainty quantification, which has been lacking in the calibration literature.

**Estimating Aleatoric Uncertainty**

There exists two common approaches in accurately estimating aleatoric uncertainty: model-based and inference-based. Model-based approaches capture aleatoric uncertainty by specifying a reasonable model for the random component. Common methods under this approach can be either 'parametric' by assuming the data arise from a known distribution family (e.g. use heterogenous Gaussian for handling heterogeneity[**?** ], use Student's t for handling heavy-tailedness [53] , use log-normal for asymmetry [57], etc), or 'nonparametric' by deploying a flexible model without explicit assumption about data distribution (e.g. infinite Gaussian mixture [48] or implicit generative models [39]). Inference-based approaches, on the other hand, improve model's quality in uncertainty quantification by modifying the inference objective with certain uncertainty metrics, which can be the coverage probability of predictive intervals [45], proper scoring rules [31, 60, 35] or the kernelized calibration errors [30]. Recognizing that the correct quantification of predictive uncertainty requires both a flexible model and also a carefully-designed inference procedure, our method combines these two approaches under a Bayesian framework. Specifically, our modeling approach is distinct from

the previous model-based methods in that our model is "semi-parametric" (i.e. the final model $F = G \circ \Phi$ is the nonparametric augmentation ($G$) of a parametric distribution family $\Phi$). Consequently, our model exhibits better convergence behavior than the other assumption-free methods (especially when $\Phi$ is specified approximately correctly), and is able to avoid model bias when the parametric assumption in $\Phi$ is violated. At the same time, our model is principled in that the model for $F = G \circ \Phi$ is designed to obey the mathamatical constrains (e.g. monotonicity, smoothness, etc) of distribution functions, and is easier to train when compared to the other nonparametric random-component models due to being an Gaussian process.

A work that is closely related to BNE is [13], which also uses flexible transformations to augment a parametric model to better estimate the aleatoric noise. There are two main differences between BNE and the two-step approach in [13]: (1) the exact type of transformation used and (2) the inference method for the ensemble parameters $\omega$. Specifically, the transformation in BNE $G_{\mathbf{x}} : \Phi(.|\mathbf{x}) \to F^*(.|\mathbf{x})$ is an operator between distribution functions, while the transformation in [13] is a diffeomorphism $\Gamma_{\mathbf{x}} : y \to y'$ acting on the response space $y, y' \in \mathcal{Y}$. As a result, the PDFs for the two models are $f(y|\mathbf{x}, \mu) = \frac{\partial}{\partial y} \Phi(\Gamma_{\mathbf{x}}(y)|\mathbf{x}, \hat{\omega}) = \phi(\Gamma_{\mathbf{x}}(y)|\mathbf{x}, \hat{\omega}) * \gamma_{\mathbf{x}}(y)$ (Dasgupta et al), and $f(y|\mathbf{x}, \mu) = \frac{\partial}{\partial y} \mathbf{G_x}(\Phi(y|\mathbf{x}, \omega)) = g_{\mathbf{x}}(\Phi(y|\mathbf{x}, \omega)) * \phi(y|\mathbf{x}, \omega)$ (this work), where $g_{\mathbf{x}} = \frac{\partial}{\partial \Phi} G_{\mathbf{x}}$ and $\gamma_{\mathbf{x}} = \frac{\partial}{\partial y} \Gamma_{\mathbf{x}}$. As shown, the two models share a similarity in that their PDFs are the product of $\phi$ (a Gaussian PDF) and the derivative of a transformation function. However, their PDFs' exact expressions are in fact different. Comparing the two models, both models are flexible in learning the data-generating $F^*$, while the construction in this work is better posed for uncertainty quantification in ensemble learning for two reasons: (1) Interpretability: $G_{\mathbf{x}}$ is a transformation that directly "fixes" the original likelihood $\Phi$, which allow us to diagnose the misspecification in original likelihood by inspecting $G_{\mathbf{x}}[\Phi] - \Phi$. This is not easy to do under the formulation of Dasgupta et al; (2) Estimation quality for the ensemble weights $\omega$. Since [13] estimates $\omega$ under the original parametric model, the resulting estimate $\hat{\omega}$ can be biased if the original likelihood is misspecified. In comparison, our model estimates $\omega$ under the flexible BNE likelihood to avoid biased inference.

## A.2 Identifying Sources of Model Error and Uncertainty

The goal of uncertainty and error decomposition is to identify the sources of uncertainty and modeling misspecification in the current system, and to quantify their impact on the model output. The standard approach to uncertainty decomposition is to attribute the model's total predictive variance to the corresponding uncertainty sources using either analytical approaches (e.g. using either the analysis of variance (ANOVA) techniques [62, 19, 6] or the law of total variance as in [15] and in this work), or Monte Carlo simulation by reverse-propagating the predictive uncertainty backward from the model output to model parameters when the base predictors are differential equations (e.g. the *reverse ensemble streamflow prediction* method in hydrological modeling [59]).

Compared to uncertainty decomposition, error decomposition (i.e. identifying the sources of model misspecification and quantify their impact on prediction) is a more difficult task. In econometrics, this problem is being studied by an ongoing body of literature in the context of time-series modeling [11, 14, 25, 40, 24, 44]. The majority of these methods follow a *parameter augmentation* approach, i.e. augmenting the possibly misspecified model components with additive / multiplicative random effects [11, 14, 25] or with auxilary latent random variables [40], then detecting the existence of model misspecification by conducting hypothesis tests for the augmentation parameters [40, 24]. The drawback of this approach is that the augmentation parameters still need to be specified correctly to capture the data's empirical distribution, therefore misspecification may still exist after augmentation, which may render the conclusions drawn from the hypothesis tests invalid. Our method improves on these approaches by nonparametrically augmenting both the model's prediction and distribution function, thereby avoid making any explicit assumption about the augmentation parameters. Furthermore, since our model explicitly quantifies structural uncertainty, our method also provides a principled framework to comprehensively assess both the magnitude and the likelihood of different types of model misspecification, as well as their impact on different aspects of the predictive distribution. These features have been lacking in the previous literature.

## B Prior Specification for $G$: Constrained Gaussian Process

Assuming modeling $G \sim GP(0, k(\mathbf{z}, \mathbf{z}'))$ with the data pair $\{y, \mathbf{z}\}$ and denote $g = \frac{\partial}{\partial \mathbf{z}} G$ the derivative of $G$, a *constrained Gaussian process* [51, 36, 34] imposes the monotonicity constraint $\mathscr{C} = \{G | g \geq 0\}$ and boundedness constraint

$\mathscr{C} = \{G | G \geq 0, 1 - G \geq 0\}$ onto $G$ by explicitly modeling the joint posterior $f(G, g | y, \mathbf{z}, \mathscr{C})$ as:

$$P(G, g | \mathscr{C}, y, \mathbf{z}) \propto P(y | G, \mathbf{z}) P(\mathscr{C} | G, g) P(G, g). \tag{1}$$

Here $P(y | G, \mathbf{z})$ is the likelihood function for $y | G(\mathbf{z})$. $P(\mathscr{C} | g)$ is the likelihood for the non-negativity constraint $\mathscr{C}$ that assigns near zero probability to the $G$'s that have negative derivative or take value outside $[0, 1]$. In this work, we consider $P(\mathscr{C} | G, g) \propto \Phi_\sigma(g) * \Phi_\sigma(G) * \Phi_\sigma(1 - G)$ where $\Phi_\sigma$ denotes the Gaussian CDF with variance $\sigma$. The variance parameter of $\Phi$ can be adjusted to specify the strength of regularization [36], which we set to $\sigma = 0.01$ so $P(\mathscr{C} | G, g)$ drops quickly to zero in regions outside the feasibility constraint. Finally, $P(G, g)$ is the joint prior for the Gaussian process $G$ and its derivative $g$. Specifically, since differentiation is a linear operator, for $G(\mathbf{z}) \sim GP(0, k(\mathbf{z}, \mathbf{z}'))$, the derivative $g(\mathbf{z}) \sim GP(0, \frac{\partial}{\partial \mathbf{z} \partial \mathbf{z}'} k(\mathbf{z}, \mathbf{z}'))$ is again a Gaussian process. Therefore, conditional on $\mathbf{z}$, $(G, g)$ is jointly multivariate Gaussian and the $P(G, g)$ takes the form [49]:

$$\begin{bmatrix} G(\mathbf{z}) \\ g(\mathbf{z}) \end{bmatrix} \sim \phi \left( \begin{bmatrix} 0 \\ 0 \end{bmatrix}, \begin{bmatrix} k(\mathbf{z}, \mathbf{z}') & \partial_{\mathbf{z}'} k(\mathbf{z}, \mathbf{z}') \\ \partial_{\mathbf{z}} k(\mathbf{z}, \mathbf{z}') & \partial_{\mathbf{z}} \partial_{\mathbf{z}'} k(\mathbf{z}, \mathbf{z}') \end{bmatrix} \right).$$

Similar to Gaussian process (GP), the *constrained Gaussian process* (CGP) prior is proper and comes with a theoretical guarantee in posterior convergence for a wide range of learning settings [34, 56].

## C Posterior Inference

### C.1 Posterior Likelihood

To ensure efficient posterior sampling, we integrate out $G$ and $\delta$ from the model posterior, and express the model compactly as $F^*(y | \mathbf{x}) = F(y | \mathbf{x}, \mu)$ with prior:

$$F | \mu \sim CGP(\Phi_\varepsilon(y | \mathbf{x}, \mu), k_G), \quad \mu | \omega \sim GP(\sum_k f_k(\mathbf{x}) \omega_k, k_\mu), \tag{2}$$

and to avoid parameter non-identifiability, we orthogonalize $k_G$ and $k_\mu$ with respect to their mean functions $\Phi_\varepsilon$ and $\sum_k f_k(\mathbf{x})$ [50, 37]. Specifically, given a kernel matrix $\mathbf{K}$ generated by the kernel function $k$, and a mean function matrix $\mathbf{F}_{N \times K}$ that corresponds to the $K$ mean functions evaluated at $N$ data points, we can compute the orthogonal projection matrix to the residual space of mean function as $\mathbf{P}_{N \times N} = \mathbf{I} - \mathbf{F}(\mathbf{F}^\top \mathbf{F})^{-1} \mathbf{F}^\top$ where $\mathbf{F}_{N \times K}$, and compute the orthogonalized kernel matrix as $\mathbf{K} = \mathbf{P} \mathbf{K} \mathbf{P}^\top$ [50].

Consequently, given observations $\mathscr{D} = \{y_i, \mathbf{x}_i\}_{i=1}^N$, denoting $\phi(. | \mathbf{m}, \mathbf{S})$ the Gaussian probability density function (PDF) with mean $\mathbf{m}$ and covariance function $\mathbf{S}$, *Bayesian Nonparametric Ensemble* (BNE)'s model posterior $p_\mathscr{D}$ is a simple product of model PDFs $f = \frac{\partial}{\partial y} F$ and Gaussian distribution functions:

$$p_\mathscr{D}(F, \mu) \propto \left[ \prod_{i=1}^N f(y_i | \mathbf{x}_i, \mu) \right] * p(f | \mu) * p(\mu | \omega) * p(\omega), \tag{3}$$

where $p(\mu | \omega)$ is the standard GP likelihood, and $p(\omega)$ is the zero-mean Gaussian likelihood $N(0, \sigma_\omega^2 \mathbf{I})$, and $p(f | \mu)$ is the CGP likelihood specified in (1), i.e.:

$$p(f | \mu) = \phi \left( F, f \middle| \begin{bmatrix} \Phi_\varepsilon(. | \mu) \\ \phi_\varepsilon(. | \mu) \end{bmatrix}, \begin{bmatrix} k & \partial_{\mathbf{z}'} k \\ \partial_{\mathbf{z}} k & \partial_{\mathbf{z}} \partial_{\mathbf{z}'} k \end{bmatrix} \right) * \Phi_\sigma(F) \Phi_\sigma(1 - F) \Phi_\sigma(f), \tag{4}$$

where $\Phi_\varepsilon$ and $\phi_\varepsilon$ are the model cumulative distribution function (CDF) and PDF of the $\delta$-agumented ensemble (2).

### C.2 Hyperparameters

BNE's hyper-parameters are the Matérn length-scale parameters $l_\delta$ and $l_G$, and the prior variances $\sigma_\omega$ and $\sigma_\varepsilon$. Consistent with the existing GP approaches, we place the inverse-Gamma priors on the $l_\delta$ and $l_G$ and the Half Normal priors on $\sigma_\omega$ and $\sigma_\varepsilon$ [? ], i.e.:

$$l_\delta \sim invGamma(\alpha_\delta, \beta_\delta) \quad l_G \sim invGamma(\alpha_G, \beta_G)$$

where $\alpha$, $\beta$ where chosen so the prior probability for $l$ falling into a desired range $[l_{lower}, l_{upper}]$ is high. In this work, we set $l_{lower} = 2$ and $l_{upper} = 10$ such that $P_{invGamma}(l \in [2, 10] | \alpha, \beta) = 0.98$ for both $l_\delta$ and $l_G$. Finally, we use the weakly informative half-Gaussian priors for the variance parameters:

$$\sigma_\omega \sim HalfNormal(0, 5) \quad \sigma_\varepsilon \sim HalfNormal(0, 5)$$

## C.3  Computation

To fully capture the model uncertainty encoded in the posterior distribution, We perform posterior inference using Hamiltonian Monte Carlo (HMC) [42], and we use the adaptive step size proposed in [1] with implementation available in TensorFlow Probability [17]. Given data $\mathscr{D}$ of size $N$, we first estimate the value of the hyper-parameters $\{l_\delta, l_G, \sigma_\omega, \sigma_\varepsilon\}$ through an empirical Bayes approach by maximizing the posterior likelihood regularized by the hyper-parameter priors specified in C.2. We then perform full MCMC with respect to the posterior likelihood (3), fixing the hyperparamters to their estimated values from the empirical Bayes procedure.

The computation complexity of sampling from BNE posterior is $O(N^3)$ due to the need of inverting GP kernel matrices. In the case of very large datasets, we consider the simple parallel MCMC scheme proposed in [32] which is dedicated to estimating posterior quantiles (i.e. the inverse CDFs). This method divides the dataset into $K$ subsets and run MCMC with respect to an adjusted posterior in parallel in each subset. As a result the computation complexity is reduced to $O(N^3/K^3)$ in the case of $K$ machines, or $O(N^3/K^2)$ in the case of a single machine.

## D  Posterior Consistency of BNE

*Proof.* Analogous to Theorem 3.2 of [56], we show Theorem 1 by invoking Theorem 2.1 of [21] which shows the posterior consistency for general Bayesian models, provided that certain conditions on the model likelihood and the prior distribution are satisfied (i.e. condition (D.1-I)-(D.1-III) in below theorem). For completeness, we state the theorem below:

**Theorem D.1** (Posterior Consistency for General Bayesian Models). *Let $\{y_i, \mathbf{x}_i\}_{i=1}^n \sim P^*(y|\mathbf{x})$, and let $P \sim \Pi_n$ be a sequence of prior probability with support on B, and $||.||$ a metric on B. Denote $N(\varepsilon, B, d)$ the $\varepsilon$-packing number for set B with respect to metric $||.||$, also denote $KL(P, P^*)$ the Kullback-Leibler (KL) divergence, and $V(P, P^*) = E^* \left( log(\frac{P}{P^*})^2 \right)$ the $L_2$ metric for log likelihood ratio between P and $P^*$.*

*Suppose that for a sequence $\varepsilon_n \to 0$ and $n\varepsilon_n^2 \to \infty$, a constant C and a sequence of set $B_n$, we have:*

$$log N(\varepsilon_n, B_n, ||.||) \leq n\varepsilon_n^2 \tag{D.1-I}$$

$$\Pi_n(B \backslash B_n) \leq exp(-n\varepsilon_n^2(C+4)) \tag{D.1-II}$$

$$\Pi_n \left( KL(P, P^*) \vee V(P, P^*) \leq \varepsilon_n^2 \right) \geq exp(-n\varepsilon_n^2 C) \tag{D.1-III}$$

*Then for sufficiently large M, we have that in probability:*

$$\Pi_n \left( d(P, P_0) \geq M\varepsilon_n \Big| \{y_i, \mathbf{x}_i\}_{i=1}^n \right) \to 0$$

Therefore to show posterior consistency, only need to show the three conditions in above theorem hold. To this end, we make use of the below theorem from [34] which derives the aforementioned conditions for CGP. We state this theorem below:

**Theorem D.2** (Conditions for Posterior Consistency in CGP). *Let g be a Borel measurable, zero-mean constrained Gaussian random element in a separable Banach space $(\mathbb{B}, ||.||_\infty)$ with reproducing kernel Hilbert space (RKHS) $(\mathscr{H}_k, ||.||_{\mathscr{H}_k})$. Define the concentration function $\psi_{g^*}(\varepsilon) = inf_{\hat{g} \in \mathscr{H}_k, ||\hat{g} - g^*||_\infty \leq \varepsilon} ||\hat{g}||_{\mathscr{H}_k}^2 - log P(||g||_\infty \leq \varepsilon)$.*

*For any number $\varepsilon_n > 0$ satisfying $\psi_{g^*}(\varepsilon_n) \leq n\varepsilon_n^2$, and any constant $C \geq 1$ with $e^{-Cn\varepsilon_n^2} < \frac{1}{2} * \frac{E(f(\mathscr{C}|g))}{E(||g||_{\mathscr{H}_k}^2)\vee 1}$, for $g^*$ a function contained in the closure of $\mathscr{H}_k$ in $\mathbb{B}$ that is $\varepsilon_n$-feasible, there exists a measurable set $B_n \subset \mathbb{B}$ such that:*

$$log N(2\varepsilon_n, B_n, ||.||) \leq 2Cn\varepsilon_n^2 \tag{D.2-I}$$

$$P(g \notin B_n) \leq e^{-Cn\varepsilon_n^2} \tag{D.2-II}$$

$$P(||g - g^*|| < 2\varepsilon_n) \geq e^{-n\varepsilon_n^2} \tag{D.2-III}$$

Consequently, to show the posterior consistency for BNE with respect to the $L_2$ metric $E^*(||F - F^*||_2)$, we only need to show the three conditions in Theorem D.1 is satisfied by making use of Theorem D.2.

Notice for sufficiently small $\varepsilon_n$, $F^*$ is $\varepsilon_n$-feasible (i.e. $E(f(\mathscr{C}|F^* + s)) > E(f(\mathscr{C}|s))$ where $s$ is a bounded random noise $s \sim GP(0, k)$ and $||s|| < \varepsilon_n$, see [34] for full definition) since the zero function $\mathbf{0}(x) = 0$ lies on the boundary of the constrain set $\mathscr{C}$, therefore we can apply Theorem D.2 for BNE. Only left to check if the three conditions in Theorem D.1 are satisfied by making use of Theorem D.2. It is easy to see that conditions D.1-I and D.1-II are satisfied by D.2-I and D.2-II. Therefore only need to show that condition D.1-III also holds. To this end, notice that by Lemma 3.2 of [56] the $KL(P, P^*)$ and $V(P, P^*)$ metrics in condition (D.1-III) is upper bounded by the $L_2$ metric (denoted as $||.||$) in equation (D.2-III) up to a multiplicative constant. Therefore D.2-III implies D.1-III. As a result, the three conditions in Theorem D.1 are satisfied, which implies posterior convergence. $\qquad\square$

# E    BNE's Predictive Distribution

**Expression for Model Predictive Mean**    Recall the "Darth Vadar rule" [41]:

$$E(s(y)|\mathbf{x}) = \int_{y \in \mathscr{Y}} \frac{\partial}{\partial y} s(y) * \left[I(y > 0) - F(y|\mathbf{x})\right] dy \tag{5}$$

Also recall that BNE's model CDF is $F(y|\mathbf{x}, G, \Phi) = G\left[F_S(y|\mathbf{x}, \Phi)\right]$. Therefore we can expresse the predictive mean $E(y|\mathbf{x}, G, \Phi)$ for full BNE in terms of its CDF:

$$
\begin{aligned}
E(y|\mathbf{x}, G, \Phi) &= \int_{y \in \mathscr{Y}} I(y > 0) - F(y|\mathbf{x}, G, \Phi) dy \\
&= \int_{y \in \mathscr{Y}} I(y > 0) - G\left[F_S(y|\mathbf{x}, \Phi)\right] dy \\
&= \int_{y \in \mathscr{Y}} \left[I(y > 0) - F_S(y|\mathbf{x}, \Phi)\right] + \left[F_S(y|\mathbf{x}, \Phi) - G\left[F_S(y|\mathbf{x}, \Phi)\right]\right] dy \\
&= \underline{\int_{y \in \mathscr{Y}} \left[I(y > 0) - F_S(y|\mathbf{x}, \Phi)\right] dy} + \int_{y \in \mathscr{Y}} \left[F_S(y|\mathbf{x}, \Phi) - G\left[F_S(y|\mathbf{x}, \Phi)\right]\right] dy
\end{aligned}
$$

Notice in the last line of above expression, the first integral (underlined) is the predictive mean with respect to the additive ensemble model $Y = \sum_{k=1}^{K} f_k(\mathbf{x})\mu_k + \delta(\mathbf{x}) + \varepsilon$. Therefore:

$$
\begin{aligned}
E(y|\mathbf{x}, G, \Phi) &= \sum_{k=1}^{K} f_k(\mathbf{x})\mu_k + \underbrace{\delta(\mathbf{x})}_{D_\delta(y|\mathbf{x})} + \underbrace{\int_{y \in \mathscr{Y}} \left[F_S(y|\mathbf{x}, \Phi) - G\left[F_S(y|\mathbf{x}, \Phi)\right]\right] dy}_{D_G(y|\mathbf{x})} \\
&= \sum_{k=1}^{K} f_k(\mathbf{x})\mu_k + D_\delta(y|\mathbf{x}) + D_G(y|\mathbf{x}) \tag{6}
\end{aligned}
$$

As shown, the predictive mean for full BNE is composed of three parts: 1) the predictive mean of the original ensemble $\sum_{k=1}^{K} f_k(\mathbf{x})\mu_k$, 2) the prediction error due to bias in prediction function $D_\delta(y|\mathbf{x}) = \delta(\mathbf{x})$, and 3) the prediction error due to bias in distribution function $D_G(y|\mathbf{x}) = \int \left[F_S(y|\mathbf{x}, \Phi) - G[F_S(y|\mathbf{x}, \Phi)]\right] dy$.

Consequently, we can assess the impact of model bias in distribution function and in distribution specification using $D_\delta$ and $D_G$:

$$D_\delta(y|\mathbf{x}) = \delta(\mathbf{x})$$

$$D_G(y|\mathbf{x}) = \int \left[ F_S(y|\mathbf{x},\Phi) - G[F_S(y|\mathbf{x},\Phi)] \right] dy,$$

and since both $\delta$ and $G$ are random variables, $D_G(y|\mathbf{x})$ and $D_\delta(y|\mathbf{x})$ are also random variables whose posterior distributions can be computed through the posterior distributions of $\delta$ and $G$.

**Expression for Predictive Distribution's Other Properties**   We can generalize the above approach further to describe the impact of the distribution biases on other properties of the predictive distribution (e.g. predictive variance, skewness, multi-modality, etc). That is, given a summary statistic $s(y)$ that describes certain property of the predictive distribution, we can assess the impact of the distribution bias on such property as

$$D_G(s(y)|\mathbf{x}) = \int \frac{\partial}{\partial y} s(y) * \left[ \Phi(y|\mathbf{x},\mu) - G[\Phi(y|\mathbf{x},\mu)] \right] dy, \tag{7}$$

and quantify the associated uncertainty using $P(D_G(s(y)|\mathbf{x}) > 0)$. For example, we can assess *predictive variance* using the variance statistic $s(y) = (y - E(y))^2$, *asymmetry* using the skewness statistic $s(y) = [(y - E(y))/SD(y)]^3$, and *multi-modality* using the kurtosis statistic $s(y) = [(y - E(y))/SD(y)]^4$ [2, 12, 61]. In Section 4 and 5, we illustrate this method in experiments and apply it to detect a real-world ensemble system's systematic bias for air pollution prediction.

To derive (7), we again use the "Darth Vadar rule" (5):

$$E(s(y)|\mathbf{x},G,\Phi) = \int_{y \in \mathscr{Y}} \frac{\partial}{\partial y} s(y) \left[ I(y > 0) - F(y|\mathbf{x},G,\Phi) \right] dy$$

$$= \underbrace{\int_{y \in \mathscr{Y}} \frac{\partial}{\partial y} s(y) \left[ I(y > 0) - F_S(y|\mathbf{x},\Phi) \right] dy}_{E(s(y)|\mathbf{x},\Phi)} + \underbrace{\int_{y \in \mathscr{Y}} \frac{\partial}{\partial y} s(y) \left[ F_S(y|\mathbf{x},\Phi) - G[F_S(y|\mathbf{x},\Phi)] \right] dy}_{D_G(s(y)|\mathbf{x})}.$$

$$= E(s(y)|\mathbf{x},\Phi) + D_G(s(y)|\mathbf{x})$$

where the first component $E(s(y)|\mathbf{x},\Phi)$ is the expected value of the summary statistics under the additive ensemble model $Y = \sum_{k=1}^K f_k(\mathbf{x})\mu_k + \delta(\mathbf{x}) + \varepsilon$, i.e. the BNE model without $G$, and the second component $D_G(s(y)|\mathbf{x})$ is the change in $E(s(y)|\mathbf{x})$ due to model bias in distribution specification.

Consequently, we can assess the impact of model bias in distribution function on $E(s(y)|\mathbf{x})$ as:

$$D_G(s(y)|\mathbf{x}) = \int_{y \in \mathscr{Y}} \frac{\partial}{\partial y} s(y) \left[ F_S(y|\mathbf{x},\Phi) - G[F_S(y|\mathbf{x},\Phi)] \right] dy$$

# F   Uncertainty Decomposition with BNE

## F.1   Structural Uncertainty Terms are Non-negative

In this section, we show that the two structural uncertainty terms in the decomposition

$$\mathscr{I}((\omega,\delta,G),y|\mathbf{x}) = \underbrace{\mathscr{I}((\omega,\delta,G),y|\mathbf{x}) - \mathscr{I}((\omega,\delta),y|\mathbf{x},G=I)}_{structural,G} +$$

$$\underbrace{\mathscr{I}((\omega,\delta),y|\mathbf{x},G=I) - \mathscr{I}(\omega,y|\mathbf{x},\delta=0,G=I)}_{structural,\delta} + \underbrace{\mathscr{I}(\omega,y|\mathbf{x},\delta=0,G=I)}_{parametric}$$

are non-negative. We show this by showing a general result that for a parameter set $\Theta$ that can be partitioned into two groups $\Theta = \{\Theta_1, \Theta_2\}$, we always have:

$$\mathscr{I}((\Theta_1,\Theta_2),y|\mathbf{x}) - \mathscr{I}(\Theta_1,y|\mathbf{x},\Theta_2=\theta_2) \geq 0 \tag{8}$$

If we can show (8), then we have shown the two structural uncertainty terms are non-negative by taking $\Theta_1 = \omega$ and $\Theta_2 = (\delta, G)$ for structural uncertainty in $\delta$, and $\Theta_1 = (\omega, \delta)$ and $\Theta_2 = G$ for structural uncertainty in $G$.

We now show (8) is true. Show this by showing below two inequalities:

$$\mathscr{I}((\Theta_1, \Theta_2), y|\mathbf{x}) \geq \mathscr{I}((\Theta_1, \Theta_2 = \theta_2), y|\mathbf{x}) \tag{9}$$

$$\mathscr{I}((\Theta_1, \Theta_2 = \theta_2), y|\mathbf{x}) \geq \mathscr{I}(\Theta_1, y|\mathbf{x}, \Theta_2 = \theta_2) \tag{10}$$

First show (9), notice that:

$$
\begin{aligned}
\mathscr{I}((\Theta_1, \Theta_2), y|\mathbf{x}) &= \int f(\Theta_1, \Theta_2, y|\mathbf{x}) log \frac{f(\Theta_1, \Theta_2, y|\mathbf{x})}{f(\Theta_1, \Theta_2|\mathbf{x})f(y|\mathbf{x})} d\Theta_1 dy \\
&= \int f(\Theta_1, \Theta_2|\mathbf{x}) * f(y|\Theta_1, \Theta_2, \mathbf{x}) log \frac{f(y|\Theta_1, \Theta_2, \mathbf{x})}{f(y|\mathbf{x})} d\Theta_1 dy \\
&= \int f(\Theta_1, \Theta_2|\mathbf{x}) * KL\Big[f(y|\Theta_1, \Theta_2, \mathbf{x})||f(y|\mathbf{x})\Big] d(\Theta_1, \Theta_2) \\
&\geq \int f(\Theta_1, \Theta_2|\mathbf{x}) * KL\Big[f(y|\Theta_1, \Theta_2, \mathbf{x})||f(y|\mathbf{x})\Big] d(\Theta_1, \Theta_2 = \theta_2) \\
&= \int f(\Theta_1, \Theta_2 = \theta_2, y|\mathbf{x}) log \frac{f(\Theta_1, \Theta_2 = \theta_2, y|\mathbf{x})}{f(\Theta_1, \Theta_2 = \theta_2|\mathbf{x})f(y|\mathbf{x})} d\Theta_1 dy \\
&= \mathscr{I}((\Theta_1, \Theta_2 = \theta_2), y|\mathbf{x})
\end{aligned}
$$

where the first inequality in above expression follows since the KL term is always non-negative. Now show (10):

$$
\begin{aligned}
\mathscr{I}((\Theta_1, \Theta_2 = \theta_2), y|\mathbf{x}) &= \int f(\Theta_1, \Theta_2 = \theta_2, y|\mathbf{x}) log \frac{f(\Theta_1, \Theta_2 = \theta_2, y|\mathbf{x})}{f(\Theta_1, \Theta_2 = \theta_2|\mathbf{x})f(y|\mathbf{x})} d\Theta_1 dy \\
&= \int f(\Theta_1, \Theta_2 = \theta_2, y|\mathbf{x}) \Big[log \frac{f(\Theta_2 = \theta_2, y|\mathbf{x})}{f(\Theta_2 = \theta_2|\mathbf{x})f(y|\mathbf{x})} + log \frac{f(\Theta_1, y|\mathbf{x}, \Theta_2 = \theta_2)}{f(\Theta_1|\mathbf{x}, \Theta_2 = \theta_2)f(y|\mathbf{x}, \Theta_2 = \theta_2)}\Big] d\Theta_1 dy \\
&= KL\Big[f(\Theta_2 = \theta_2, y|\mathbf{x})||f(\Theta_2 = \theta_2|\mathbf{x})f(y|\mathbf{x})\Big] + \mathscr{I}(\Theta_1, y|\mathbf{x}, \Theta_2 = \theta_2) \\
&\geq \mathscr{I}(\Theta_1, y|\mathbf{x}, \Theta_2 = \theta_2),
\end{aligned}
$$

and the inequality follows since the KL divergence is always non-negative.

Finally, by combining (9) and (10), we have shown (8).

# G  Experiments

## G.1  Data Generation Mechanism and Computation Environment

We sample $x$ from a mixture of Gaussians $\{N(-4,0.4),N(0,1),N(4,0.4)\}$, and sample $y$ from $y = 7*sin(x)+3*cos(\frac{x}{2})\varepsilon$ with $\varepsilon \sim Weibull(\alpha,\beta=1)$. We set $\alpha = 3*exp(-|x|)$ so the noise distribution is highly skewed and heavy-tailed at the margin of input space $x \in \mathscr{X}$ but is symmetric (i.e. low skewness) in the middle. All computation is done on a Intel Core i7-6700HQ machine with Nvidia GeForce GTX 1070 GPU and 16 Gb RAM. Implementation is done in TensorFlow Probability under Ubuntu 14.04 LTS [17].

## G.2  Visualization of Base Models

For base models $\{f_k\}_{k=1}^K$, we pre-train 3 kernel ridge regression models on half of the data points, using periodic and radial basis function (RBF) kernels as their kernel families (see Figure G.1b for an example). We then train the ensemble model using the remaining half of the data points.

(a)

(b)

Figure G.1: **Left**: data generation mechanism; **Right**: base model predictions (fitted with 50 training data points).

## G.3  Further Description of Experiment Results

**Uncertainty-aware Model Bias Detection**  We quantify the impact of the original ensemble's model biases on model's predictive distribution in Figure 3d and 3e. Recall as discussed in Section 3.1, such impact are quantified by the random quantity $D_\delta$ and $D_G$, respectively, and model's posterior confidence in such impact are described by the probabilities $P(D_\delta > 0)$ and $P(D_G > 0)$. Figure 3d shows model's posterior confidence in original ensemble's predictive bias due to misspecification in prediction function (color indicates the direction of the bias). As shown, the high-confidence regions in 3d precisely reflected regions in 3a where model bias exists and data is available. Specifically, the confidence is high if the model bias is severe or there are enough data to justify the existence of bias, and is low otherwise. Figure 3e shows model's posterior confidence in Bayesian additive ensemble's bias in predictive mean (upper plot) and variance (lower plot) due to misspecification in the distribution function. As shown, the high-confidence regions in the upper plot of Figure 3e precisely captured regions where biases in the predictive mean exist (see Figure 3b). Specifically, notice that in regions where the data is sufficient (e.g. $x \in [1.0, 1.5]$), the posterior confidence is able to identify biases in predictive mean even if their magnitude is small. Finally, in the lower plot of Figure 3e, model's posterior confidence in variance bias accurately captured regions where the additive ensemble model fail to account for the decreased variance in data's empirical distribution on the left and right margin of the feature space $x \in \mathscr{X}$.

# H  Application

## H.1  Background

To assess exposure in air pollution health studies, many research groups are developing distinct spatial models (exposure models) to predict ambient air pollution concentrations even in areas where air pollution monitors are sparse. These

different models have different inputs (including remote sensing, chemical transport models, and land use variables) and employ different algorithms (e.g., generalized additive models, random forests and neural networks). As a result, the model predictions from base models differ across space; if these were to be integrated in an ensemble model, thus, the variability among the base model predictions would drive the predictive uncertainty (e.g., see Figure H.2). Consequently, for the purpose of model validation and refinement, it is important to identify the spatial regions where model bias exists, and how disagreement across candidate models may significantly impact the ensemble's predictive uncertainty.

Figure H.2: Visualization of 2011 $PM_{2.5}$ predictions from different base models in Eastern Massachussets, USA. **Left** [18], **Middle**: [28], **Right**: [16]

## H.2 Results

| Model | BNE | BME | BAE | stack |
|---|---|---|---|---|
| loo RMSE | $0.762 \pm 0.09$ | $0.833 \pm 0.07$ | $1.077 \pm 0.13$ | $1.472 \pm 0.15$ |

Table H.1: Mean and Standard Deviation for leave-one-out cross-validation RMSE in the 2011 PM$_{2.5}$ ensemble prediction.

Figure H.3: Posterior confidence in original model's prediction bias, i.e. $P(D_\delta(y|\mathbf{x}) > 0)$. Blue/Red color indicates evidence of over-/under-estimation