[Reviews · NeurIPS 2019]

Reviewer 1



The authors should discuss connections of the density regression approach in Section 5 of http://www3.stat.sinica.edu.tw/ss_newpaper/SS-2018-0231_na.pdf. The cited paper operates on the conditional density instead of the cdf by expressing f(y | x, \mu) = phi( \gamma(y); \mu(x), \sigma^2) \gamma(y) where \gamma is assumed to be a diffeomorphism. I think the two models are inherently very similar as the cdf transform that the authors proposed can be linked to \gamma. I do think there are certain advantages of operating on the density rather than on the cdf, the primary reason being computational convenience. The constrained Gaussian process that the authors propose seems to be complicated to implement; the HMC approach that the authors alluded to is not automated requiring to choose carefully the step sizes. Moreover, no comparisons are provided with the fully flexible classical methods. The authors should at least consider some of the off the shelf conditional density estimators, e.g. the np-package as a point of comparison. Also, flexibility in the mean function is not discussed, does the approach have a natural extension to include moderately high dimensional predictors?

Reviewer 2



The motivation for this work stems from the fact that ensemble models with individual models in the ensemble given, and only their weights to be estimated, may not fit the data well. The contributions of this paper can be used to improve an ensemble model and enable assessing the shortcomings of the ensemble. Outside the data, the model reduces to the original ensemble, while close to the data the bias term and the correction to the model’s noise distribution may significantly affect the output. The elements of the solution have been existing before, and their application to improve ensemble models seems to be the main novel innovation of the article. The topic seems well motivated and could be of interest to people trying to apply and understand ensemble models in practice. The theoretical derivations are interesting and establish a solid foundation for the choices made, although somewhat straightforward. A strength of the article is that it is well-written and mathematical details are presented clearly and accurately (also the Supplement). A weakness in presentation is that the quality of figures is clearly insufficient (legends, labels etc. are very difficult if not impossible to read in many cases). ---AFTER REBUTTAL AND DISCUSSION--- I thank the authors for the detailed rebuttal. To summarize, the contribution of the article is to show how to make an ensemble model more flexible, by introducing a residual process for bias correction and a transformation of the noise distribution. Furthermore, the impact of different sources of uncertainty can be estimated, and this is likely to be useful when interpreting the ensemble's predictions in practice (as demonstrated in the Application section). This is a sensible and interesting contribution, and potentially useful to the community, although somewhat straightforward. Novelty of the work and its relation to the Dasgupta paper is a concern: a transformation of the noise distribution seems to be presented in both. The rebuttal argues that the present work has benefits in terms of interpretability and accuracy of the ensemble weights, but this would require more treatment (than is possible in a rebuttal) to be fully convincing. The works appear to be done independently of each other. A strength of the paper is the clarity of mathematical detail. On the other hand, the clarity of figures is not sufficient for NeurIPS, and I hope the authors invest the effort to improve the figures if the paper gets accepted. Acknowledging the concerns in presentation and novelty, I still think the paper is acceptable but borderline, and stick to my original score 6.

Reviewer 3



### 1) Content: ### The paper proposes a Bayesian Nonparametric Ensemble (BNE) approach that augments an existing (deterministic) ensemble with nonparametric models to account for different sources of uncertainty in the data. The augmented model is specifically designed to allow for the disambiguation between aleatoric and epistemic uncertainty as well as between different sources of epistemic uncertainty (parametric vs. structural). The model design is well-motivated and described in great detail with additional details and derivation included in the appendix. Additionally, the authors provide a proof that the BNE posterior distribution will concentrate around the true data-generating distribution as the number of training samples is being increased. The paper also includes an illustrative toy example and an empirical comparison between BNE and related models. ### 2) Empirical evaluation: ### The empirical evaluation is insightful, and the plots are clear and complement the main text well. However, the empirical evaluation is also very limited in its scope. I am convinced that BNE outperforms the three other methods on the toy dataset as well on the PM2.5 exposure dataset, but note that BNE is by construction more expressive than BAE. Furthermore, we note that the RMSE results included in the Appendix H2 show that the loo RMSEs of BNE and BME are within one anothers standard deviations. The empirical comparison could be made more convincing (see under improvements below). That being said, the empirical evaluation presented in the paper is limited but, within its scope, thorough and insightful, and I believe it is borderline sufficient. I would urge the authors to extend the empirical comparisons before publication in line with the suggested improvements below. ### 3) Presentation ### The paper is superbly written. The presentation is clean and the writing is extremely clear. The submission is accompanied by a detailed appendix with easy-to-follow and detailed derivations and additional experimental results. The plots could be improved (see under improvements below).

[Author Response · NeurIPS 2019]

We thank all reviewers for taking the time to provide detailed feedback and valuable suggestions for our work. We
address the reviewers' detailed comments below:
• **Comparison with Dasgupta et al. (2017)**. We thank Reviewer # 1 for pointing out this interesting work. Both our
model and Dasgupta et al. (2017) use flexible transformations to augment a parametric model, but the two approaches
differ in (1) the exact type of transformation used and (2) the inference method for the ensemble parameters $\omega$.
Specifically, the transformation in our method $G_{\mathbf{x}} : \Phi(.|\mathbf{x}) \to F^*(.|\mathbf{x})$ is an operator between distribution functions,
while the transformation in Dasgupta et al. (2017) is a diffeomorphism $\Gamma_{\mathbf{x}} : y \to y'$ acting on the response space
$y, y' \in \mathcal{Y}$. As a result, the PDFs for the two models are $f(y|\mathbf{x}, \mu) = \frac{\partial}{\partial y} \Phi(\mathbf{\Gamma}_{\mathbf{x}}(y)|\mathbf{x}, \hat{\boldsymbol{\omega}}) = \phi(\mathbf{\Gamma}_{\mathbf{x}}(y)|\mathbf{x}, \hat{\boldsymbol{\omega}}) * \boldsymbol{\gamma}_{\mathbf{x}}(y)$
(Dasgupta et al), and $f(y|\mathbf{x}, \mu) = \frac{\partial}{\partial y} \mathbf{G}_{\mathbf{x}}(\Phi(y|\mathbf{x}, \boldsymbol{\omega})) = \boldsymbol{g}_{\mathbf{x}}(\Phi(y|\mathbf{x}, \boldsymbol{\omega})) * \phi(y|\mathbf{x}, \boldsymbol{\omega})$ (this work), where $g_{\mathbf{x}} = \frac{\partial}{\partial \Phi} G_{\mathbf{x}}$
and $\gamma_{\mathbf{x}} = \frac{\partial}{\partial y} \Gamma_{\mathbf{x}}$. As shown, the two models share a similarity in that their PDFs are the product of $\phi$ (a Gaussian
PDF) and the derivative of a transformation function. However, their PDFs' exact expressions are in fact different.
Comparing the two models, both models are flexible in learning the data-generating $F^*$. However the construction
in this work is better posed for uncertainty quantification in ensemble learning for two reasons: (1) Interpretability:
$G_{\mathbf{x}}$ is a transformation that directly "fixes" the original likelihood $\Phi$, which allow us to diagnose the misspecification
in original likelihood by inspecting $G_{\mathbf{x}}[\Phi] - \Phi$. This is not easy to do under the formulation of Dasgupta et al; (2)
Estimation quality for the ensemble weights $\omega$. Since Dasgupta et al. (2017) estimates $\omega$ under the original parametric
model, the resulting estimate $\hat{\omega}$ can be biased if the original likelihood is misspecified. In comparison, our model
estimates $\omega$ under the flexible BNE likelihood to avoid biased inference. In the final manuscript, we will include above
discussion in the Related Work section (at the end of Section A.1).
• **Implementation difficulty of CGP and HMC.** Since CGP is simply a GP model with probit-based likelihood
penalties, implementing CGP is in fact not difficult. Any GP model can be converted to a CGP model by adding log
probit terms to the GP log likelihood (e.g., in TensorFlow Probability, this can be done in one line as `gp_likelihood +`
`tfd.Normal.log_cdf(g)`) and then apply MCMC as before. The HMC method proposed in this work in fact does not
require hand tuning. It uses an automated adaptive step size scheme that is readily available in TensorFlow Probability
(see Appendix Section C.3). Alternatively, one can also use the *No U-Turn Sampler* (NUTS) implemented in Stan.
• **Flexibility in the mean function.** As shown in equation (5) of the main text, BNE's mean function consists of the
original ensemble, the residual process $\delta$ and a bias correction term due to $\mathbf{G}$. In addition to the base predictors, the
flexibility of BNE's mean function is mainly driven by the residual process $\delta$, and domain experts can select a flexible
kernel for $\delta$ to best approximate the data-generating function of interest. (e.g., a RBF kernel to approximate arbitrary
continuous functions over a compact support (Micchelli et al., 2006) ). In the manuscript, we will include this discussion
when first introducing the residual process (line 103-106 of the main text)
• **Extension to moderately high dimensional predictors.** The BNE framework can be naturally extended to high
dimension by choosing kernel functions for $\delta$ and $\mathbf{G}$ that are suitable for high-dimensional problems. Example choices
include the additive kernel (Durrande et al., 2011) or (deep) neural network kernel (Bach, 2014; Lee et al., 2017).
Alternatively, one could also build variable selection into the model using shrinkage priors such as the Automatic
Relevance Determination (ARD), spike-and-slab, or Horseshoe (Bobb et al., 2015; Vo and Pati, 2017). In the final
manuscript, we will include the above discussion in the Conclusion and Future Work section.
• **In the experiment, BNE is by construction more expressive than BAE.** We thank Reviewer # 3 for highlighting
an important part of our experiment design. Indeed, BAE is an abalated version of BNE (i.e. without $\mathbf{G}$). The goals to
include BAE are to see (1) if $\mathbf{G}$ leads to significant improvement in large sample sizes, and (2) if $\mathbf{G}$ severely overfits
data and leads to worse performance in small sample sizes. Figures 4-5 suggest that the former is true, but not the latter.
• **In Appendix H2, the low RMSEs of BNE and BME are within one another's standard deviations.** We thank
Reviewer # 3 for making this observation. Indeed, this result is expected and is consistent with what we observed in the
simulation experiment (Figures 4-5): In a small sample (where uncertainty is high), BNE is competitive with BME in
prediction performance, while providing several advantages when the goal is uncertainty quantification (i.e., uncertainty
decomposition in Figure 5, and model diagnosis in Figure H.3).
• **Extend empirical comparison.** Following suggestions by Review # 1 and # 3, in the final manuscript we will extend
the empirical comparisons in Figure 4-5 to include two more models: the popular Deep Ensemble (Lakshminarayanan
et al., 2017) and the classic `npcdist` from the np package. Deep Ensemble and `npcdist` fits a finite Gaussian mixture
and a kernel smoother to the data, respectively. In terms of performance, Deep Ensemble is expected to perform
similarly to BME which also uses a Gaussian mixture. `npcdist` is expected to perform similarly to BNE in this 1D
experiment, however generalizing kernel density estimators to higher dimensions is usually more difficult (Scott, 2015).
We feel the two scalable GP approaches cannot be directly compared to BNE since they still rely on a parametric model
likelihood from a known distribution family. However, we do note that BNE can be made scalable by estimating $\mathbf{G}$ and
$\delta$ using the variational inducing point method in Hensman et al. (2015). However, the disadvantage of this approach is
that the uncertainty estimate may be inaccurate since the variational family usually does not fully capture the posterior
distribution. Research into scalable inference method that provides good uncertainty quantification is an important
future direction of this work. We will include the above discussion in the Conclusion and Future Work section.

[Meta-Review · NeurIPS 2019]

The authors consider the problem of ensemble learning. They augment a traditional ensemble with one Gaussian process (GP) to address prediction bias and another (constrained, monotonic) GP to address miscalibration. They use this augmented ensemble to separate aleatoric and epistemic uncertainty. They demonstrate the method on simulated and real data and provide supporting theory. The reviewers agree the approach is clear, straightforward, and practically useful. In forming their revision, the authors should be sure to address a number of concerns that arose during review, rebuttal, and discussion. In particular, the authors have promised in their rebuttal to address the issues of improved empirical comparisons with existing methods and to more carefully discuss some existing work that was missed in the first draft. In addition to making minor plot edits to bring plots in line with NeurIPS formatting guidelines, I strongly encourage the authors to improve their explanations and discussions of the plots; make sure that every item (e.g. line, shaded area, etc) that is plotted is fully explained to the reader.